# The Removal Efficiencies of Several Temperate Tree Species at Adsorbing Airborne Particulate Matter in Urban Forests and Roadsides

**Myeong Ja Kwak [1]**, **Jongkyu Lee [1]**, **Handong Kim [1]**, **Sanghee Park [1]**, **Yeaji Lim [1]**, **Ji Eun Kim [1]**, **Saeng Geul Baek [2]**, **Se Myeong Seo [3]**, **Kyeong Nam Kim [3]** and **Su Young Woo [1],***

[1] Department of Environmental Horticulture, University of Seoul, Seoul 02504, Korea; 016na8349@hanmail.net (M.J.K.); gpl90@naver.com (J.L.); blasterkhd92@gmail.com (H.K.); parksanghee0930@gmail.com (S.P.); oxll2l@naver.com (Y.L.); amarg@naver.com (J.E.K.)
[2] National Baekdudaegan Arboretum, Bonghwa 36209, Korea; bsg1175@bdna.or.kr
[3] Korea Forestry Promotion Institute, Seoul 07570, Korea; ssm9911@kofpi.or.kr (S.M.S.); uforest81@kofpi.or.kr (K.N.K.)
* Correspondence: wsy@uos.ac.kr; Tel.: +82-10-3802-5242

**Abstract:** Although urban trees are proposed as comparatively economical and eco-efficient biofilters for treating atmospheric particulate matter (PM) by the temporary capture and retention of PM particles, the PM removal effect and its main mechanism still remain largely uncertain. Thus, an understanding of the removal efficiencies of individual leaves that adsorb and retain airborne PM, particularly in the sustainable planning of multifunctional green infrastructure, should be preceded by an assessment of the leaf microstructures of widespread species in urban forests. We determined the differences between trees in regard to their ability to adsorb PM based on the unique leaf microstructures and leaf area index (LAI) reflecting their overall ability by upscaling from leaf scale to canopy scale. The micro-morphological characteristics of adaxial and abaxial leaf surfaces directly affected the PM trapping efficiency. Specifically, leaf surfaces with grooves and trichomes showed a higher ability to retain PM as compared to leaves without epidermal hairs or with dynamic water repellency. *Zelkova serrata* (Thunb.) Makino was found to have significantly higher benefits with regard to adsorbing and retaining PM compared to other species. Evergreen needle-leaved species could be a more sustainable manner to retain PM in winter and spring. The interspecies variability of the PM adsorption efficiency was upscaled from leaf scale to canopy scale based on the LAI, showing that tree species with higher canopy density were more effective in removing PM. In conclusion, if urban trees are used as a means to improve air quality in limited open spaces for urban greening programs, it is important to predominantly select a tree species that can maximize the ability to capture PM by having higher canopy density and leaf grooves or trichomes.

**Keywords:** adsorption; leaf surfaces; microstructure; particulate matter; roadsides; urban forests

## 1. Introduction

Urban forests have a wide range of benefits, particularly the purification of air and water quality, mitigation of urban heat islands [1,2], space services for recreational activities [3], and enhancement of the physical and mental health of urban dwellers [4,5]. In recent years, air pollution, including airborne particulate pollutants in urban areas, has become a serious problem in developed and developing countries. It has a detrimental effect on humans, as well as on living organisms, plants, and environments [6]. The ability of plants to absorb and metabolize gaseous atmospheric pollutants and nanoparticles has been reported in previous studies. Thus, in recent years, the air purification

service of urban trees has become increasingly important, as they adsorb atmospheric particulate matter (PM), linking adaptation, mitigation, and sustainable management to air pollutants.

The PM is a major atmospheric pollutant with aerodynamic particle diameters in the range of 0.001–100 μm and is typically monitored as coarse particulate matter ($PM_{10}$), less than 10 μm, and fine particulate matter ($PM_{2.5}$), less than 2.5 μm, in aerodynamic diameter [7]. PM is widely recognized as one of the pollutants that are most harmful to natural and human health and is estimated to be responsible for approximately 8 million premature deaths annually worldwide. Since the International Agency for Research on Cancer (IARC) designated $PM_{2.5}$ as a Group 1 carcinogen to humans in 2013 [8], the demand for diversification of urban air quality management, including airborne particulates, is increasing in order to enhance human health and environmental welfare.

Trees within urban life zones are one of the key elements in mitigating and reforming atmospheric environmental issues, but many planters do not take into account the potential of urban policy and strategy. Improving particle deposition rates on plant leaf surfaces can reduce the concentration of suspended PM near the surface. Therefore, plants have the functional advantage of reducing the effects of air pollutant exposure on human health in urban areas; this is one of the recognized ecosystem services of urban vegetation [9,10]. Compared to other surfaces in the city, plants, especially coniferous trees, can improve the adsorption and deposition of fine dust particles owing to the finely divided structure of their leaves. Conifers have a larger collection surface per unit area and reduced laminar boundary layer, which limits particle adsorption [11].

Studies are underway to investigate the adsorption and/or absorption of PM by plant species [10,12–18]. Nevertheless, there is a lack of knowledge about PM deposition on the leaves of urban trees. Given the limited area of urban green spaces, there is a need to select the most effective plant species to mitigate PM for urban greening. Although many studies have focused on quantifying the amount of PM adsorbed on plant leaves, there is still a knowledge gap about the potential capacities for PM accumulation on the leaf surface among plant species.

Plants adsorb PM on their leaf surfaces and absorb gaseous pollutants through the leaf pores, thereby directly removing air pollutants and improving air quality. The ability of trees to adsorb and retain PM air pollutants depends on several factors such as the canopy type, leaf and branch density, and leaf cuticular micromorphology (e.g., grooves, trichomes, and wax). In addition, there are important interactions between local meteorological conditions and ambient PM concentrations [18–20].

Given the difficulty of reducing airborne PM in the short term, it is necessary to select plants with high efficiency at removing the particulate air pollution in urban environments and to reflect them in air quality improvement policies. Since the leaf surface characteristics of plants are known to reflect the ability of plants to adsorb and retain PM [21], the ability of plants to remove suspended particulate pollutants can vary among species. Depending on the amount of PM mass deposited on the leaf surface of trees, tree species might vary their own critical physiological, biochemical, morphological, and architectural features and growth [22–24].

The primary aim of this study was therefore to demonstrate the amount of PM adsorbed by major tree species, especially species that are commonly planted in both urban forests and roadsides, and to evaluate tree species that are highly effective at capturing PM particles from the atmosphere in temperate zone trees. The second aim was to determine whether PM can cause physiological and biochemical deterioration when tree species are exposed to ambient PM levels in urban green spaces.

## 2. Materials and Methods

### 2.1. Site Description

This study was performed to detect the potential capacity of several tree species for adsorbing particulate pollutants on their leaf surfaces. Leaf sampling and monitoring of trees used in the field experiment were conducted in the Seoul Forest Park and Yangjae Citizen's Forest (hereafter denoted as SFP and YCF, respectively) on the northern and southern sides of Seoul, South Korea (Figure 1).

These areas appropriately represent urban air quality in South Korea. The SFP and YCF, which are surrounded by roadsides, are 480,994 $m^2$ (32°32.6′ N, 127°2.4′ E) and 258,991 $m^2$ (37°28.2′ N, 127°2.2′ E), respectively, and are managed forests mainly comprising Korean red pine (*Pinus densiflora* Siebold & Zucc.), Korean flowering cherry (*Prunus yedoensis* Matsum.), sawleaf zelkova (*Zelkova serrata* (Thunb.) Makino), American sycamore (*Platanus occidentalis* L.), and maidenhair tree (*Ginkgo biloba* L.). We hereinafter used the terms inside (SFP-IN, YCF-IN) and roadsides (SFP-OUT, YCF-OUT) of the Seoul Forest Park (SFP) and Yangjae Citizen's Forest (YCF), respectively.

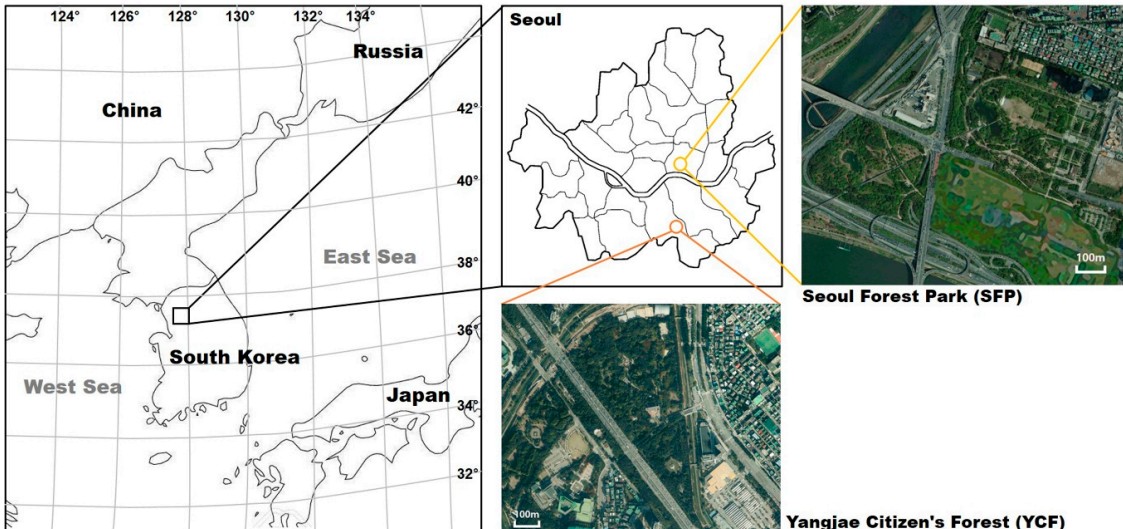

**Figure 1.** Location of two sampling sites in Seoul, South Korea: Seoul Forest Park (SFP) and Yangjae Citizen's Forest (YCF). National Geographic Information Institute.

Data on the average $PM_{10}$ and $PM_{2.5}$ concentrations for the past five years were obtained from the Seoul Metropolitan Government. Based on the annual average concentrations, the $PM_{10}$ and $PM_{2.5}$ concentrations at the two sites were similar to those at Seoul (Table 1).

**Table 1.** Annual averages of airborne $PM_{10}$ and $PM_{2.5}$ concentrations in Seoul, Seongdong District, and Seocho District during a 5-year period (2013–2017).

| Site | $PM_{10}$ | $PM_{2.5}$ |
|---|---|---|
| Seoul | 45.4 [1] | 24.6 |
| Seongdong District [2] | 47.4 | 24.4 |
| Seocho District [2] | 47.8 | 24.1 |

[1] Annual average concentrations of airborne $PM_{10}$ and $PM_{2.5}$ particles in Seoul; [2] Two districts in Seoul, Seongdong District and Seocho District, are local government districts located on the northern and southern sides of Seoul and located in Seoul Forest Park (SFP) and Yangjae Citizen's Forest (YCF), respectively.

*2.2. Data Collection*

In the present study, we monitored and tested the potential adsorption capacity of PM particles throughout the growing season on the leaf surface of five representative species that frequently occur in the urban forests and roadsides of Seoul (see Section 2.3 for more details). The most commonly planted tree species in the Seoul area's living zones are *G. biloba*, *P. occidentalis*, *Z. serrata*, *P. yedoensis*, and *P. densiflora*, based on the public data service [25]. For each detected tree species on the inside and outside of the two urban forests, nine sample trees for each species were selected, and the measurements of tree diameter and height were determined using a digital dendrometer (Criterion™ RD1000, Laser Technology, USA) with the aid of a laser distance meter (Leica DISTO™ A5; Leica Geosystems, Heerbrugg, Switzerland).

Prior to the analysis procedure, all leaf samples were cut into branches, carefully performed so that the number of particulates on the leaf surfaces was not affected. Leaf samples to determine PM adsorption capacity and air pollution tolerance index (APTI) were picked at a tree height of 3 to 6 m and obtained from branches at the outer part of the canopy exposed to the atmosphere. The samples were also collected three times during four consecutive months (June, July, August, and September of 2018) from each site. Three to five branches were cut from selected each tree and then placed directly in individually labeled paper bags. Leaf samples for ascorbic acid and pH analysis were detached from the branches, immediately packed with aluminum foil to avoid contamination, frozen in liquid nitrogen, and thus stored in deep freezer at −80°C until biochemical analysis. Leaf samples for relative moisture content and chlorophyll analysis were stored in an icebox. After sample processing, all samples collected from the sites were immediately transferred to the laboratory. In addition, the concentration data on individual airborne particulates including the total suspended particulate matter were directly obtained 72 times in both urban forests and roadsides using a DustMate handheld PM monitor (Turnkey Co. Ltd., British) over a three-month field study at each site.

## 2.3. PM Particles Deposited on a Unit Leaf Area Basis (ULA)

The adsorption of PM was measured by modifying the methods of [26] and [27]. Five tree species as mentioned above were tested for their capacity to capture airborne particulates through their leaf surfaces. To recover the air-suspended particulates captured on leaf surfaces, leaf samples (as mentioned in Section 2.2) of each species were washed sequentially by water cleaning and ultrasonic cleaning. First, twelve leaves of each species (seventy samples in the case of pine needles) were washed by immersing them in an individual beaker filled with 270 mL of deionized water and then stirred for 10 min using a shaker. Second, the beaker containing leaves washed with deionized water was put directly into an ultrasonic cleaning machine and homogeneously cleaned by an ultrasonic wave of 500W for 1 min. After the leaves were ultrasonically washed, 270 mL of the obtained eluent was dispensed, with 90 mL put into each of three beakers. We also measured their leaf area after washing, using WinFOLIA PRO 2013a software (Regent Instruments Inc., Quebec, Canada). Before the experiment, all test beakers were thoroughly dried after ultrasonic cleaning and weighed using an electronic balance (W1). Each beaker containing eluting solvent was uniformly covered with a clean filter paper to prevent the pollution of other particulates. Next, all beakers were dried for 3 days at 70 °C using an oven dryer until the moisture completely evaporated, cooled in the balance chamber, equilibrating the temperature and humidity, and immediately weighed again using an electronic balance (W2). The PM mass filtered through each washing step was calculated as W2–W1 and represented as the masses of particulates per ULA (mg·cm$^{-2}$).

## 2.4. Quantifying the Overall PM Removal Capacity Including Leaf Area Index (LAI) by Different Tree Species

The leaves of different plants have different surface areas and are distributed differently depending on the space. The total amount of PM adsorption on leaves of different tree species may vary depending on the leaf surface area available for PM capture [28]. Because the high LAI value corresponding to a very dense canopy is an important factor in PM deposition based on scaled up ecosystem scale from individual leaf level deposits, we hypothesized that the adsorption of particulates on leaf surfaces would be equivalent to different parts of the tree [29].

We developed an LAI-based method for estimating the amount of PM adsorbed in green leaf area per unit ground surface. The LAI was measured using an LAI-2000 Plant Canopy Analyzer (Li-Cor). After the LAI measurement, expressed as the ratio of the leaf area sum per unit land area (m$^2$·m$^{-2}$), the PM-capturing capacity of different tree species (mg·cm$^{-2}$) was calculated by multiplying the LAI value by the PM adsorption amount per ULA values.

Allometric equations are most frequently used to estimate the total leaf area (TLA) based on the diameter at breast height (DBH) representing tree species from different environments. The TLA values from the DBH of trees at each monitoring site were calculated using the prediction equation model as

shown in Table 2 [30–32]. The PM-capturing capacity of different tree species (mg·cm$^{-2}$) based on the TLA using the prediction equation model was calculated by multiplying the TLA value by the amount of PM adsorption per ULA values.

**Table 2.** Allometric equations to calculate TLA values using DBH for tree species.

| Species | Leaf Morphology | Allometric Equation (m$^2$·tree$^{-1}$) | $R^2$ | References |
|---|---|---|---|---|
| *P. densiflora* | EC [1] | y = 0.2988 × (DBH$^2$) − 7.5336 × (DBH) + 74.075 | 0.94 | [30] |
| *Z. serrata* | DC [2] | y = EXP(4.033) × EXP((0.045 × DBH) −1) × EXP(0.12706/2) | 0.91 | [31] |
| *P. occidentalis* | DC | y = EXP(5.198) × EXP((0.021 × DBH) −1) × EXP(0.23508/2) | 0.74 | [31] |
| *P. yedoensis* | DC | y = 3.036 × (EXP(0.09 × DBH) −1) | 0.80 | [32] |
| *G. biloba* | DC | y = EXP(3.410) × EXP((0.053 × DBH) −1) × EXP(0.30207/2) | 0.86 | [31] |

[1] Ec: evergreen coniferous species; [2] Dc: deciduous broad-leaved species.

### 2.5. Calculation of Air Pollution Tolerance Index (APTI)

APTI was adopted to assess their tolerance level to air pollutants based on the below four parameters. APTI as a marker to evaluate plant tolerance to air pollutants has been evaluated from four physiological and biochemical parameters: leaf extract pH (pH), relative water content (RWC), total chlorophyll (TChl), and ascorbic acid (AsA) [6,33,34]. For each of these parameters, AsA serves as an important coenzyme in multiple biological metabolism reactions, TChl acts as one of the main essential parts of energy production in plants, and directly related to the health status of plants based on stress environmental conditions, RWC is a useful indicator for the performance of cell protoplasmic permeability, and intracellular pH regulation is required for trafficking network of proteins and transporting small molecule such as hormones [33,34].

In order to measure the degree of susceptibility to air pollution in each tree species, three fully mature leaves were randomly selected and collected. Leaf samples were immediately put in a plastic bag and stored separately in an icebox or in liquid nitrogen. The analysis of each parameter for APTI was performed as previously described [6,33–36]. APTI values were calculated using the following formula:

$$APTI = (A \times (T + P) + R)/10 \tag{1}$$

where $A$ is the AsA (mg·g$^{-1}$ FW, [35]), $T$ is the TChl (mg·g$^{-1}$ FW, [6]), $P$ is the leaf extract pH [34], and $R$ is the RWC (%, [36]).

### 2.6. Statistical Analysis

Statistical analysis of all data was performed using IBM SPSS Statistics 25 software (SPSS Inc., IBM Company Headquarters, Chicago, IL, USA). One-way analysis of variance (ANOVA) was used to examine statistically significant differences in the PM adsorption based on ULA, LAI, and TLA values on five tree species in urban forests and roadsides. Then, ANOVA was performed to identify statistically significant differences between groups, and Duncan's multiple range test was used to test the significance of differences between groups. The statistical significant differences between urban forests and roadsides within each site of SFP and YCF were performed using a paired-samples *t*-test.

## 3. Results

### 3.1. Comparison of Particulate Matter between Urban Forests and Roadsides

According to the surveyed annual average of PM$_{10}$ and PM$_{2.5}$ concentrations in Seoul for the last five years [37], the average PM$_{10}$ and PM$_{2.5}$ of Seoul were 45.4 and 24.6 µg·m$^{-3}$, the average PM$_{10}$ and PM$_{2.5}$ of Seongdong-gu were 47.4 and 24.4 µg·m$^{-3}$, and the average PM$_{10}$ and PM$_{2.5}$ of Seocho-gu were 47.8 and 24.1 µg·m$^{-3}$, respectively (Table 1). The maximum concentration of PM$_{2.5}$ in 2013 was 94 µg·m$^{-3}$, and the number of PM$_{2.5}$ warning alerts issued was one per year. On the other hand, the maximum concentration of PM$_{2.5}$ in 2017 was 157 µg·m$^{-3}$, and the number of alerts issued about

$PM_{2.5}$ was five per year (data not shown). This reflects the recent trend toward increased issuance of "warning" notifications about $PM_{2.5}$ concentrations, which are highly toxic to humans in urban areas. As shown in Figure 2, the ambient PM concentrations of urban forests and roadsides during the study period were found to be lower compared to the Korean air quality 24 h average standard (i.e., 100 $\mu g \cdot m^{-3}$ for $PM_{10}$ and 35 $\mu g \cdot m^{-3}$ for $PM_{2.5}$), especially the concentrations of PM in urban forests, which were considerably lower than those of roadsides. The PM particles of roadsides at sites of SFP and YCF were approximately 2-fold higher in TSP, $PM_{10}$, $PM_{2.5}$, and $PM_{1.0}$ compared to urban forests, while these particles of urban forests showed the levels of decreased pollutants of almost 7% to 69% at urban forests than at roadsides (Figure 2).

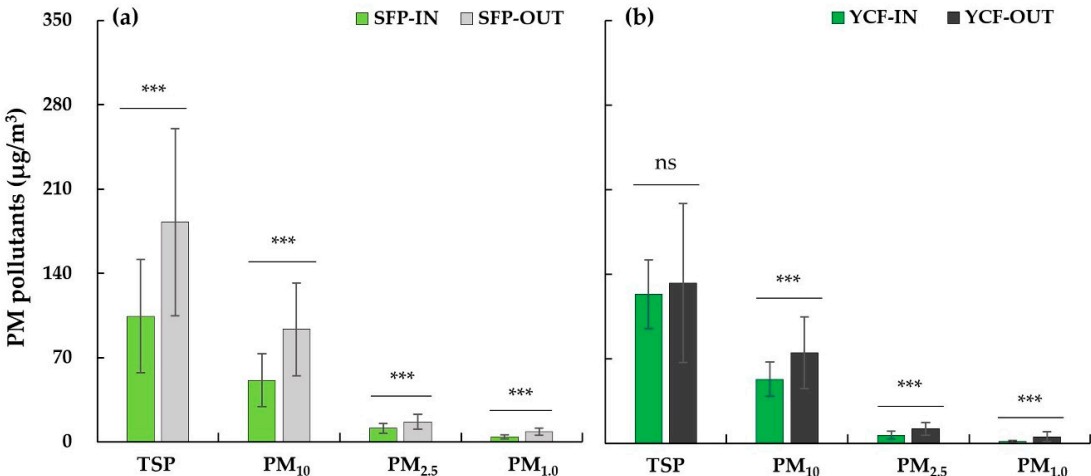

**Figure 2.** Comparison of particle concentrations on PM pollutants including TSP, $PM_{10}$, $PM_{2.5}$, and $PM_{1.0}$ at urban forests and roadsides of (**a**) Seoul Forest Park (SFP-IN, SFP-OUT) and (**b**) Yangjae Citizen's Forest (YCF-IN, YCF-OUT). Each bar represents the mean of 72 replicates. Error bars refer to the standard deviation (SD) of the mean. Light green bars, SFP-IN; grey bars, SFP-OUT; green bars, YCF-IN; black bars, YCF-OUT. Asterisks represent significant differences between urban forests and roadsides (i.e., SFP-IN and SFP-OUT; YCF-IN and YCF-OUT) within each site (paired *t*-test, * $p < 0.05$; ** $p < 0.01$; *** $p < 0.001$; ns, not significant).

### *3.2. PM Adsorption of a ULA Basis on Different Tree Species*

The present study was focused on estimating the potential capacity of five major street trees in Seoul (South Korea) to adsorb suspended particulate pollutants. The foliar PM deposition of major species in urban areas varied depending on the plant species. There were also significant differences in particulate adhesion on a per ULA basis. These data revealed that, in general, *Z. serrata* was more effective at adsorbing PM in both urban forests and roadsides of two distinct sites of SFP and YCF, as compared to other tree species. In more detail, *Z. serrata* adsorbed 0.05 to 0.14 $mg \cdot cm^{-2}$, owing to leaf surfaces that were densely covered hairy trichomes (Figure 5a–c), whereas *G. biloba* showed the least adsorption capacity (0.01 $mg \cdot cm^{-2}$), and expressed a sustained water repellency during the growth season (data not shown). As shown in Figure 3, the results indicated that there was no statistically significant difference in the particulate adhesion between YCF-IN and YCF-OUT for each species. However, *Z. serrata* showed statistical difference exists in particulate adhesion per ULA between SFP-IN and SFP-OUT. There were significant variations in PM adsorption capacities among tree species, and the capacities of species with a maximum efficiency of PM adsorption were about 2.5 to 6 times higher as compared to species with a minimum efficiency. With regard to the capacity for adsorption of particulate pollutants on a per ULA basis (Figure 3a), *Z. serrata* was the highest, followed by *P. occidentalis*, *P. densiflora*, *P. yedoensis*, and *G. biloba* ($p < 0.001$).

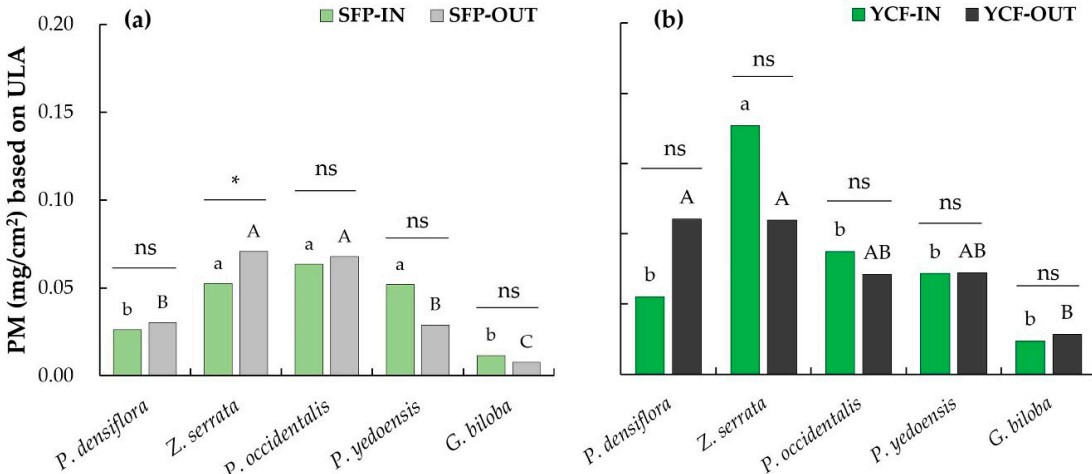

**Figure 3.** Comparison of the capacity to capture airborne PM particles on a per unit leaf area basis (ULA) of five tree species in urban forests and roadsides of (**a**) Seoul Forest Park (SFP-IN, SFP-OUT) and (**b**) Yangjae Citizen's Forest (YCF-IN, YCF-OUT). Each bar represents the mean of nine replicates. Light green bars, SFP-IN; grey bars, SFP-OUT; green bars, YCF-IN; black bars, YCF-OUT. Different lowercase letters mean significant differences among tree species within each SFP-IN and YCF-IN, while different uppercase letters mean significant differences among tree species within each SFP-OUT and YCF-OUT (Duncan's multiple range after one-way ANOVA). Asterisks represent significant differences between urban forests and roadsides (i.e., SFP-IN and SFP-OUT; YCF-IN and YCF-OUT) within each site (paired *t*-test, * $p < 0.05$; ** $p < 0.01$; *** $p < 0.001$; ns, not significant).

### 3.3. PM Adsorption Based on LAI and TLA of Different Tree Species.

The amount of PM adsorption ($g \cdot m^{-2}$) in both urban forests and roadsides recalculated based on the LAI is shown in Figure 4. Measurements of PM adsorption based on LAI showed no statistically significant differences between urban forests and roadsides at YCF site and were concordant with a result of the PM adsorption based on ULA (Figure 4b). In contrast, the PM adsorption based on LAI of *Z. serrata* and *P. occidentalis* in SFP site exhibited a maximum absolute difference between urban forests and roadsides (Figure 4a). Based on PM adsorption calculated by LAI, *Z. serrata* had the highest PM capture capacity (1.4 to 3.0 $g \cdot m^{-2}$), while *G. biloba* had the lowest capture capacity (0.1 to 0.4 $g \cdot m^{-2}$), compared to other species (one-way ANOVA, $p < 0.001$), *P. occidentalis* (0.8 $g \cdot m^{-2}$), *P. densiflora* (0.4 to 1.2 $g \cdot m^{-2}$), and *P. yedoensis* (0.3 to 0.7 $g \cdot m^{-2}$) at two sites of SFP and YCF (Figure 4). The TLA values of each species, determined by using the prediction equation model, was in the following order: *P. occidentalis*, *P. densiflora*, *Z. serrata*, *G. biloba*, and *P. yedoensis* (data not shown). In these PM adsorption based on the TLA values, *Z. serrata* showed the highest amount of PM adsorption (37~96 $g \cdot tree^{-1}$), with an average of 66.6 $g \cdot tree^{-1}$ in the range of DBH, followed by *P. yedoensis* (45.3 $g \cdot tree^{-1}$), *P. densiflora* (24.2 $g \cdot tree^{-1}$), and *G. biloba* (10.7 $g \cdot tree^{-1}$).

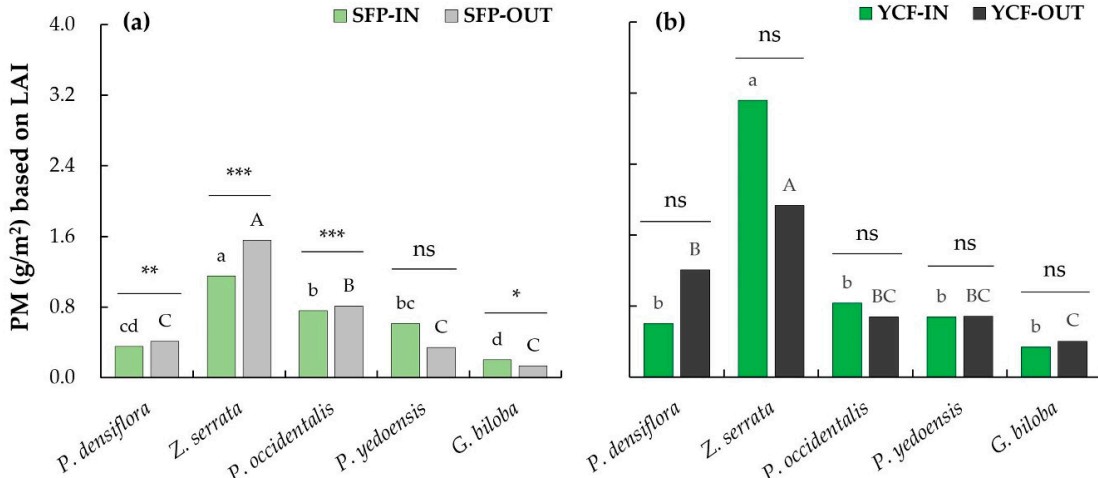

**Figure 4.** The amount of PM adsorption based on LAI of five tree species in urban forests and roadsides of (**a**) Seoul Forest Park (SFP-IN, SFP-OUT) and (**b**) Yangjae Citizen's Forest (YCF-IN, YCF-OUT). Each bar represents the mean of nine replicates. Light green bars, SFP-IN; grey bars, SFP-OUT; green bars, YCF-IN; black bars, YCF-OUT. Different lowercase letters mean significant differences among tree species within each SFP-IN and YCF-IN, while different uppercase letters mean significant differences among tree species within each SFP-OUT and YCF-OUT (Duncan's multiple range after one-way ANOVA). Asterisks represent significant differences between urban forests and roadsides (i.e., SFP-IN and SFP-OUT; YCF-IN and YCF-OUT) within each site (paired *t*-test, * $p < 0.05$; ** $p < 0.01$; *** $p < 0.001$; ns, not significant).

### 3.4. Suitable Biomonitors through Calculation of the Air Pollution Tolerance Index (APTI) from Biochemical Parameters

As shown in Table 3, the APTI was used to rank tolerance to air pollution. The APTI value of various tree species was determined by analyzing four major biochemical parameters; namely, AsA, TChl, pH, and RWC (see Supplementary Files for details: Tables S1 and S2). At site SFP, the results indicated that there was no statistically significant difference in APTI values between SFP-IN and SFP-OUT for each species during the study period. The mean APTI values of each tree species at SFP-IN and SFP-OUT were as follows (in ascending order): *P. occidentalis* (9.3, 9.0), *P. densiflora* (8.9, 8.8), *P. yedoensis* (8.6, 8.7), *G. biloba* (8.2, 8.4), and *Z. serrata* (8.0, 8.4). The APTI values of five tree species in YCF-IN and YCF-OUT at site YCF were high in *P. occidentalis* (9.0, 8.7) and *P. densiflora* (8.7, 8.9) (Table 3). Interestingly, *Z. serrata* showed a statistically significant difference in APTI values between YCF-IN and YCF-OUT, as compared to other tree species. Nonetheless, the APTI values of *Z. serrata* were slightly lower than those of *P. occidentalis*, *P. densiflora*, and *P. yedoensis* in both urban forests and roadsides at two sites of SFP and YCF.

**Table 3.** Assessment of air pollution tolerance index (APTI) on selected five tree species in urban forests (SFP-IN, YCF-IN) and roadsides (SFP-OUT, YCF-OUT) at Seoul Forest Park (SFP) and Yangjae Citizen's Forest (YCF).

|  | Sites | *P. densiflora* | *Z. serrata* | *P. occidentalis* | *P. yedoensis* | *G. biloba* |
|---|---|---|---|---|---|---|
| **SFP** | **SFP-IN** | 8.9 ± 0.5 [ab] | 8.0 ± 0.7 [c] | 9.3 ± 0.4 [a] | 8.6 ± 0.7 [b] | 8.2 ± 0.3 [c] |
|  | SFP-OUT | 8.8 ± 0.4 [A] | 8.4 ± 0.5 [B] | 9.0 ± 0.5 [A] | 8.7 ± 0.4 [A] | 8.4 ± 0.3 [B] |
| YCF | YCF-IN | 8.7 ± 0.5 [a] | 6.9 ± 0.7 [c***] | 9.0 ± 0.7 [a] | 8.7 ± 0.6 [a] | 8.2 ± 0.4 [b] |
|  | YCF-OUT | 8.9 ± 0.6 [A] | 8.1 ± 0.6 [C] | 8.7 ± 0.5 [AB] | 8.5 ± 0.5 [B] | 8.1 ± 0.5 [C] |

Values are expressed as the mean ± SD (*n* = 15). Different superscript letters denote significant differences between values within the same row (*p* < 0.05). Asterisks represent significant differences between urban forests and roadsides (i.e., YCF-IN and YCF-OUT) within each site (paired *t*-test, *** *p* < 0.001).

*3.5. Specificity of PM according to Leaf Micromorphological Structures*

There were clear differences in the topology of leaf abaxial and adaxial surfaces on five tree species (Figure 5). Airborne particulate pollutants typically accumulated around grooves and epidermal trichomes of adaxial surfaces (especially *Z. serrata*) and embedded in stomata of abaxial surfaces, blocking the entrance of the stomata (Figure 5a–c). In the case of a detailed survey of epidermal trichomes, the leaf micromorphology of *Z. serrata* was noticeably more complex on the adaxial surfaces owing to their hairy trichomes (Figure 5a). Both *Z. serrata* (Figure 5a,c) and *P. yedoensis* (Figure 5d,f) showed prominently higher levels of PM particles on the adaxial surfaces. In general, the larger leaf roughness was observed on the leaf adaxial surfaces (especially *P. yedoensis*, Figure 5f). Moreover, *P. occidentalis* adsorbed PM particles on grooves with slightly rough surfaces on the adaxial surfaces (Figure 5g,i). Interestingly, *G. biloba* markedly embedded airborne PM particles inside the surrounding stomata of the abaxial surfaces (Figure 5k), and there were a few particulate pollutants on the sparsely arranged grooves of the adaxial sides (Figure 5j,l). On the other hand, *P. densiflora* typically appeared to have the ability to accumulate high airborne particulate particles by exhibiting wax crystals with a dense arrangement in surrounding leaf abaxial and adaxial surfaces (Figure 5m–o). We also analyzed the elemental composition of various types of particles (<10 μm) on leaf surfaces using a field emission scanning electron microscopy/energy-dispersive X-ray spectrometer (FESEM/EDX). The EDX spectra indicated that the airborne particulates on leaf surfaces contained a major elemental composition of Si, Al, Fe, and Mg, suggesting that these crystals are mostly aluminosilicate/silica mineral and Si–Al rich fly ash (data not shown).

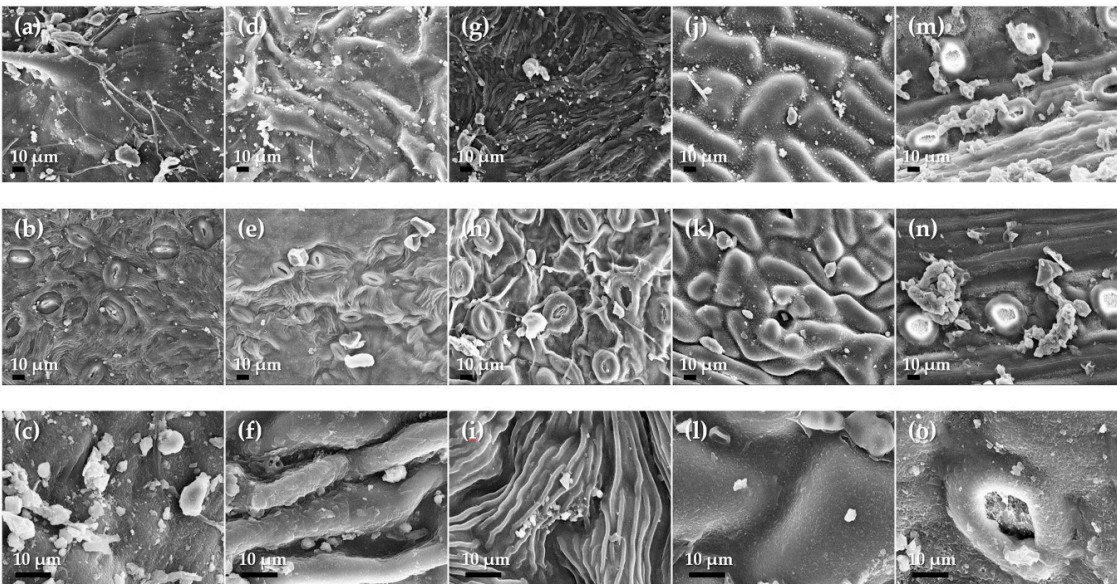

**Figure 5.** Variability of surface topography of adaxial leaf surfaces showing grooves and trichomes (**a**,**d**,**g**,**j**,**m**,**c**,**f**,**I**,**l**,**o**) and abaxial leaf surfaces showing stomata (**b**,**e**,**h**,**k**,**n**). (**a**–**c**) *Z. serrata*, (**d**–**f**) *P. yedoensis*, (**g**–**i**) *P. occidentalis*, (**j**–**l**) *G. biloba*, and (**m**–**o**) *P. densiflora*. Note PM particle accumulation in (**c**,**f**,**o**). Airborne particulate pollutants typically accumulated around grooves and epidermal trichomes of adaxial leaf surfaces and embedded in guard cells surrounding open stomata of abaxial leaf surfaces. Scale bars = 10 μm.

## 4. Discussion

There were significant variations in the PM adsorption capacities of the different tree species. A temperate deciduous broad-leaved tree (Ulmaceae), *Z. serrata*, exhibited the highest capacity of 0.06 to 0.11 mg·cm$^{-2}$ on PM capture per unit leaf area. Because plants generally exhibit a species-specificity in associated responses to air pollution, the efficiency for PM adsorption varies across plant species.

The leaf micro-morphological features are known to directly affect the agglomeration and capture of fine particles by trees. More specifically, in previous studies that reported a possible association between specific leaf characteristics and PM adsorption, leaf microstructures such as stomata, grooves, and trichomes were found to be optimal areas for capturing PM particles owing to their coarse and adhesive properties [18,38–40]. Notably, our results indicate that the leaf surfaces with high grooves or trichomes have a markedly enhanced ability to retain PM (especially $PM_{2.5}$) as compared to smooth leaf surfaces. Furthermore, leaf shape and venation of the broad-leaved species have no significant influence on the immobilization rates and retention times of the fine PM, because individual leaves do not reflect the physical properties of canopy density [18].

On the other hand, the effect of leaf area on PM accumulation was observed to be dominant over other investigated characteristics (leaf size and macro- and micro-morphology). The canopy density affects air turbulence around leaves, which has been proposed as an important explanatory factor for the deposition of PM particles, enhancing dry deposition of PM on leaves. Several studies have shown that tree species with higher LAI and smaller leaf size were most effective in adsorption of PM [28,29]. LAI can be used as a surrogate for the photosynthetically active area between the atmosphere and underlying land surfaces [30]. Our current work agrees that in addition to leaf micromorphological characters, total leaf area also plays important role in capturing airborne particulates when total leaf area was incorporated in foliar PM capture efficiency for upscaling of unit leaf level to tree scale (Figures 3 and 4). Tree species (especially in *Z. serrata*) with higher LAI and smaller leaves were the most effective for PM adsorption. Therefore, the ability to adsorb and retain PM per unit leaf area is important, but it is also important to consider the total leaf area and LAI.

Remarkably, the effect of PM adsorption on unit leaf area was lower in *P. occidentalis*, *P. densiflora*, and *P. yedoensis* as compared to *Z. serrata*. The current issues and future challenges regarding airborne particulate pollutants depend largely on the tolerance and susceptibility of plants to air pollution [41]. At the physiological and biochemical level, the response of plants to air pollutants can be understood by analyzing the factors that determine tolerance and susceptibility. Many studies, even field survey research, have suggested that the air pollution tolerance index (APTI) is one of the impressive indices in evaluating the response of tolerance to air pollution [33,34,42]. Most of those results calculated the APTI using the parameters indicated in this paper such as ascorbic acid, total chlorophyll, leaf extract pH, and relative water content. These parameters act as an important coenzyme, the main essential parts of energy production, intracellular regulation for trafficking network of proteins, and a useful indicator of cell protoplasmic permeability, respectively. Previous studies have shown that ascorbic acid content, total chlorophyll content, leaf extract pH, and leaf relative water content are biochemical variables related to tolerance and susceptibility to air pollutants [6,33,34].

As shown in the assessment of the above mentioned APTI, *P. occidentalis*, *P. densiflora*, and *P. yedoensis* showed relatively high APTI values, while *Z. serrata* displayed the lowest APTI value. Tree species with higher APTI showed the reduction for PM trapping effects and tree species with lower APTI revealed a capacity to enhance the capture of air-suspended particles. Interestingly, *Z. serrata* revealed the distinguishable capacity to prolong APTI values on roadsides as compared to urban forests (Table 3), possibly by enhancing RWC (Tables S1 and S2). The higher RWC value acts as an indicator useful of the plant protoplasmic permeability, indicating possibly more tolerant of environmental pollutions. Air pollutants increase cell permeability, which eventually causes water and nutrient losses, resulting in early leaf senescence [33,34]. Thus, by enhancing RWC positively regulates their variability to regulate the physiological balance of plants under stress conditions [6]. Therefore, higher leaf RWC values are possibly more tolerant of pollutants. As shown in Figure 5, the adsorption and deposition of PM particles were observed around stomatal pores on abaxial leaf surfaces. If the leaf transpiration interrupts due to stomatal limitation by air pollutants, plants can lead to an inhibition of their photosynthesis and growth due to loss of its engine that pulls water up from the soil into the roots.

In the PM adsorption based on the TLA, *P. occidentalis* showed the highest PM adsorption amount of 95~104 g·tree$^{-1}$ among tree species due to relatively high DBH as compared to the other species.

Nevertheless, *P. occidentalis* and *G. biloba* have been reported to disperse pollen, the main airborne allergen [43] in urban trees in Seoul, Korea. The main pollen allergens (Pla a 1, putative invertase inhibitor; Pla a 2, polygalacturonase; Pla a 3, nsLTP) of *Platanus acerifolia* have been reported [44–46]. The major allergen, Pla a 1, from *P. acerifolia* pollen, has also been reported in *P. occidentalis* [47,48]. In Seoul, *G. biloba*. which has the highest planting rate owing to its climate resilience and resistance to disease and insect pests, has recently required improvement in the selection of species owing to its foul-smelling fruits and the allergic reactions caused by its inhalant allergen contents [49]. Therefore, *P. occidentalis* and *G. biloba* may be unsuitable as tree species for superior PM removal efficiency because they are likely to cause allergies during the spring season and are to be replaced with other species owing to various problems. Interestingly, the hydrophobic leaf surface with dense wax crystals reduces the interfacial area that can help to effectively adsorb PM particles, leading to the reduction of trapping potential in particulate pollutants [13]. Nevertheless, PM particles permanently encapsulated in epicuticular wax layers can be expected to maintain particles encapsulated regardless of widespread rainfall conditions, suggesting that leaf microstructure and wax layer could be important factors to maximize PM trapping effects.

Across previous studies [18,19,38,50], there is clear and consistent evidence of the effects of the roughness of leaf surfaces and trichome densities of adaxial and abaxial leaf surfaces in determining PM retention capacity. The ability of urban vegetation to remove pollutants is well known [51], but there is a lack of information on the leaf microstructures such as grooves and trichomes of various tree species. Moreover, the PM particles on leaf surfaces showed a strong positive correlation with abaxial trichomes, thereby potentially improving PM adsorption. Results also showed the positive relationships between PM removal and leaf surface groove ratio, whereas stomatal density showed less association in capturing PM (Figure S1). As mentioned in Figure 5, leaf microstructures such as the grooves and trichomes of *Z. serrata* are believed to improve its ability to capture and retain PM particles as compared to other species. Furthermore, the air purification abilities of trees to adsorb and retain particulate pollutants can be affected by various factors, including stem, branch, canopy type, leaf area, and especially leaf microstructures (grooves, trichomes, glands, and epicuticular wax layer), showing significant species specificity [18–20].

Interestingly, *G. biloba* has been found to decrease PM accumulation by self-cleaning its leaf surfaces owing to its water repellency; therefore, airborne PM particles could be sufficiently embedded in the surrounding stomata of abaxial leaf surfaces. Obviously, sustained water repellency leads to the repeated removal of PM from leaves by different types of precipitation, thereby preventing successful deposition of suspended particles on leaf surfaces during the whole growing season [50]. Additionally, the leaf water repellency exhibited by some *G. biloba* species can purify the leaf surfaces by aiding in PM removal during rainy and foggy weather [52]. Therefore, the leaves of *G. biloba*, which maintain water repellency over the whole leaf lifetime, appear to be well protected from permanent deposition as well as damage caused by particulate pollutants. In addition, these tree species have been reported to be very resistant to environmental pollution [50,53].

Canopy density, PM concentration, particle size distribution, and wind speed are important factors that can potentially affect particle deposition on trees [54]. When wind carrying $PM_{2.5}$ crosses leaves, the boundary layers are relatively fixed and form a barrier between leaf surfaces and the ambient air. Recent evidence of PM adsorption by interspecific plant leaves has shown that coniferous trees have a higher $PM_{2.5}$ adsorption capacity as compared to broadleaved trees in urban environments [18,40,55]. The thin boundary layer of long and narrow needle leaves is more conducive to the deposition of $PM_{2.5}$ on the leaf surface and has a high ability to retain fine particles because it does not affect $PM_{2.5}$ release during rainfall. Therefore, coniferous tree species have the clear benefit of excellent PM abatement in urban areas, especially during the winter and early spring seasons, when there are no leaves on broadleaved trees in temperate climatic zones.

On the other hand, pine trees are highly susceptible to pollutants, and thus are not recommended for use in areas with high levels of gaseous pollutants. For example, ozone ($O_3$) can have a negative

impact on the net photosynthetic capacity of pine leaves and greatly reduces dry weight [56]; sulfur dioxide ($SO_2$) causes necrosis of pine leaves [57]; and atmospheric sulfur and nitrogen depositions cause nutritional imbalances and suppress or accelerate growth patterns in Scots pines [58]. In addition, nitrogen dioxide ($NO_2$) causes physiological disturbance and shortens the life span of pine trees [59].

In other words, the response of tree species to ambient particulate loading is important in order to estimate the overall greening effect on potential PM abatement. Although the PM adsorption amount of conifers per unit leaf area is almost two times higher than that of broadleaf leaves, their adsorption efficiency may be only one-third of the total leaf level owing to air pollution stress. Thus, broadleaved trees can be more effective at trapping airborne PM particles at the stand level [18]. Valuable elements of urban forests, including forest area, forest structure, and leaf shape, determine the PM adsorption capacity of trees. The PM adsorption/deposition/retention capacity of plants depends on leaf structure, tree height and canopy, and source location, as well as meteorological factors [13,17]. Branches and leaves of trees and shrubs have a higher PM removal effect owing to their larger surface area and complex structure as compared to herbaceous plants. The PM reduction process in plants is caused by dry or wet deposition processes and chemical reactions in the atmosphere, vegetation, and soil [60]. In addition, the PM adsorption process is further influenced by environmental conditions, altitude, wind speed and direction, precipitation, and season and accumulation period [61]. Furthermore, annual variations in plant leaf phenology in urban areas can maximize PM trapping effects.

## 5. Conclusions

Levels of airborne particulates, especially PM, are relatively low in urban forests as compared to that on roadsides, reflecting the removal efficiencies as natural biofilters of urban trees that improve air quality. PM adsorption efficiency by trees under stressful urban conditions is different depending on leaf morphology, physical properties, and microstructure of broadleaved surfaces with well-developed trichomes and grooves. The micro-morphological characteristics such as leaf trichomes and grooves, rather than the macro-morphological characteristics of broadleaved species, can be used as effective indicators for adsorbing PM adsorption. However, their overall ability to capture PM particles by upscaling leaf level to ecosystem-scale especially emphasizes the importance of reflecting the total leaf area and LAI. Regarding the characteristics of leaves, evergreen pine needles may display a higher potential for particulate reduction, especially during winter and early spring, as compared to broad-leaved species. Therefore, as urban open space for green infrastructure programs is limited, if plants are to be used as a means to improve air quality, further studies are needed on various mitigation techniques to maximize airborne PM uptake by trees. Establishing effective policies and management controls to reduce air pollution of particulates is one of the policy tasks of the Korean government. In addition, the Seoul Metropolitan Government and the Korea Forest Service are conducting projects and research for the selection of suitable street trees and urban trees in the construction of urban forests and to establish effective policies to reduce particulate matter. This study is a field study conducted as part of green infrastructure programs. The Seoul Metropolitan Government is trying to replace street trees with trees which are able to maximize the ability to capture PM based on the scientific research data.

**Supplementary Materials:** The following are available online at http://www.mdpi.com/1999-4907/10/11/960/s1, Figure S1: Relationships between PM adsorption on leaf surfaces ($\mu g\ cm^{-2}$) and leaf micro-morphological characters of trichome density (**a**), groove area ratio (**b**), and stomatal density (**c**) in five urban tree species in Seoul, South Korea., Table S1: Biochemical parameters of selected tree species in urban forests (SFP-IN) and roadsides (SFP-OUT) at site SFP during the study period., Table S2: Biochemical parameters of selected tree species in urban forests (YCF-IN) and roadsides (YCF-OUT) at site YCF during the study period.

**Author Contributions:** All authors contributed extensively to the present study. Authors discussed the results and commented on the manuscript at all stages. S.Y.W. as a corresponding author, developed the conceptualization and funding acquisition, edited the manuscript, and supervised the present study. M.J.K. designed the present study, analyzed the data, and wrote the manuscript. J.L. developed analytical tools and administered the project. H.K., S.P., Y.L., and J.E.K. assembled in data input and analysis. S.G.B. performed the image analyses of scanning electron microscopy. S.M.S. and K.N.K. reviewed the manuscript.

**Funding:** This research was funded by Basic Science Research Program through the National Research Foundation of Korea (NRF), grant number No. 2018R1D1A1A02044683.

**Acknowledgments:** This research was supported by Basic Science Research Program through the National Research Foundation of Korea (NRF) funded by the Ministry of Education (No. 2018R1D1A1A02044683).

**Conflicts of Interest:** The authors declare no conflict of interest. The funders had no role in the design of the study; in the collection, analyses, or interpretation of data; in the writing of the manuscript, or in the decision to publish the results.

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
