# Peer review of "The Removal Efficiencies of Several Temperate Tree Species at Adsorbing Airborne Particulate Matter in Urban Forests and Roadsides"

_forests, doi:10.3390/f10110960_

Round 1

Reviewer 1 Report

Dear authors,

The paper needs some more detailed descriptions about the methods. Find comments and corrections within the paper in form of "comments".

Author Response

Author's Reply: We are thankful for pointing these comments out. In accordance with your suggestion in the Reviewer Report, we corrected and rewrote ‘Abstract’, ‘Introduction’, ‘Materials and methods’, ‘Results’, ‘Discussion’, and ‘Conclusions’ sections including your comments. Please refer to the amended and updated versions highlighted in yellow on the uploaded manuscript.

Review Report: How many leaves did you collect? What was the position within the canopy (inner/outer part) and with respect to the cardinal points? Are the leaf positions representative for the method used for calculating PM trapping capacity?

Author's Reply: Based on the mentioned the Reviewer #1's report, we have added this information in the ‘Materials and methods’ section.

Materials and Methods (lines 109-132)

2.2. Data collection

In the present study, we monitored and tested the potential adsorption capacity of PM particles throughout the growing season on the leaf surface of five representative species that frequently occur in the urban forests and roadsides of Seoul. The most commonly planted tree species in the Seoul area's living zones are G. biloba, P. occidentalis, Z. serrata, P. yedoensis, and P. densiflora, based on the public data service [25]. For each detected tree species on the inside and outside of the two urban forests, nine sample trees for each species were selected, and the measurements of tree diameter and height were determined using a digital dendrometer (Criterion™ RD1000, Laser Technology, USA) with the aid of a laser distance meter (Leica DISTO™ A5; Leica Geosystems, Heerbrugg, Switzerland).

Prior to the analysis procedure, all leaf samples were cut into branches and carefully performed so that the number of particulates on the leaf surfaces was not affected. Leaf samples to determine PM adsorption capacity and air pollution tolerance index (APTI) were picked at a tree height of 3 to 6 m and obtained from branches at the outer part of the canopy exposed to atmosphere. The samples were also collected three times during four consecutive months (June, July, August, and September of 2018) from each site. Three to five branches were cut from selected each tree and then placed directly in individually labeled paper bags. Leaf samples for ascorbic acid and pH analysis were detached from the branches, immediately packed with aluminum foil to avoid contamination, frozen in liquid nitrogen, and thus stored in deep freezer at –80°C until biochemical analysis. Leaf samples for relative moisture content and chlorophyll analysis were stored in an icebox. After sample processing, all samples collected from the sites were immediately transferred to the laboratory. In addition, the concentration data on individual airborne particulates including the total suspended particulate matter were directly obtained 72 times in both urban forests and roadsides using a DustMate handheld PM monitor (Turnkey Co. Ltd, British) over a three-month field study at each site.

Review Report: Is this the same sample described above (line 119)? If not, you should give more quantitative and qualitative information about sampling: leaves are randomly taken, but give at least the number and the position of the leaves in the canopy (exposure sides, canopy height etc.). The application of the allometric equations without this information would be bias.

Author's Reply: We have revised the sentence as follow (lines 134-138):

The adsorption of PM was measured by modifying the methods of [26] and [27]. Five tree species as mentioned above were tested for their capacity to capture airborne particulates through their leaf surfaces. To recover the air-suspended particulates captured on leaf surfaces, leaf samples (as mentioned in Section 2.2) of each species were washed sequentially by water cleaning and ultrasonic cleaning.

Review Report: 3 days

Author's Reply: We have revised our manuscript accordingly (line 146).

Each beaker containing eluting solvent was uniformly covered with a clean filter paper to prevent the pollution of other particulates. Next, all beakers were dried for 3 days at 70 °C using an oven dryer until the moisture completely evaporated, cooled in the balance chamber, equilibrating the temperature and humidity, and immediately weighed again using an electronic balance (W2). The PM mass filtered through each washing step was calculated as W2 − W1 and represented as the masses of particulates per ULA (mg·cm-2).

Review Report: Explain better the following points: if these models consider any management practice such as pruning and how LAI is used for your purpose.

Author's Reply: We have modified the ‘Materials and methods’ and ‘Discussion’ sections as follow:

‘Materials and methods’ section (lines 153-156)

2.4. Quantifying the overall PM removal capacity including leaf area index (LAI) by different tree species

The leaves of different plants have different surface areas and are distributed differently depending on the space. The total amount of PM adsorption on leaves of different tree species may vary depending on the leaf surface area available for PM capture [28]. Because the high LAI value corresponding to a very dense canopy is an important factor in PM deposition based on scaled up ecosystem scale from individual leaf level deposits, we hypothesized that the adsorption of particulates on leaf surfaces would be equivalent to different parts of the tree [29].

‘Discussion’ section (lines 364-384)

Because plants generally exhibit a species‐specificity in associated responses to air pollution, the efficiency for PM adsorption varies across plant species. The leaf micro-morphological features are known to directly affect the agglomeration and capture of fine particles by trees. More specifically, in previous studies that reported a possible association between specific leaf characteristics and PM adsorption, leaf microstructures such as stomata, grooves, and trichomes were found to be optimal areas for capturing PM particles owing to their coarse and adhesive properties [18,46-48]. Notably, our results indicate that the leaf surfaces with high grooves or trichomes have a markedly enhanced ability to retain PM (especially PM2.5) as compared to smooth leaf surfaces. Furthermore, leaf shape and venation of the broad-leaved species have no a significant influence on the immobilization rates and retention times of the fine PM, because individual leaves do not reflect the physical properties of canopy density [18].

On the other hand, canopy density affects air turbulence around leaves, which has been proposed as an important explanatory factor for the deposition of PM particles, enhancing dry deposition of PM on leaves. Several studies have shown that tree species with higher LAI and smaller leaf size were most effective in adsorption of PM [28,29]. LAI can be used as a surrogate for the photosynthetically active area between the atmosphere and underlying land surfaces [30]. In the present study, there was significant interspecies variability of their showing their overall ability to capture airborne particulates when LAI was incorporated for upscaling of unit leaf level to the ecosystem scale. Tree species with higher LAI and smaller leaves were the most effective for PM adsorption. Therefore, the ability to adsorb and retain PM per unit leaf area is important, but it is also important to consider the total leaf area and LAI.

Review Report: You should describe the methods for determining these parameters.

Author's Reply: We have added this information in the ‘Materials and methods’ section.

‘Materials and methods’ section (lines 172-180)

2.5. Calculation of air pollution tolerance index (APTI)

APTI was adopted to assess their tolerance level to air pollutants based on the below four parameters. APTI as a marker to evaluate plant tolerance to air pollutants has been evaluated from four physiological and biochemical parameters, i.e. leaf extract pH (pH), relative water content (RWC), total chlorophyll (TChl), and ascorbic acid (AsA) [6,33,34]. For each of these parameters, AsA serve as an important coenzyme in multiple biological metabolism reactions, TChl acts as one of the main essential parts of energy production in plants, and directly related to the health status of plants based on stress environmental conditions, RWC is a useful indicator for the performance of cell protoplasmic permeability, and intracellular pH regulation is required for trafficking network of proteins and transporting small molecule such as hormones [33,34].

Review Report: REF

Author's Reply: We revised the ‘Materials and methods’ section including this part as follows:

‘Materials and methods’ section (line 184)

2.5. Calculation of air pollution tolerance index (APTI)

APTI was adopted to assess their tolerance level to air pollutants based on the below four parameters. APTI as a marker to evaluate plant tolerance to air pollutants has been evaluated from four physiological and biochemical parameters, i.e. leaf extract pH (pH), relative water content (RWC), total chlorophyll (TChl), and ascorbic acid (AsA) [6,33,34]. For each of these parameters, AsA serve as an important coenzyme in multiple biological metabolism reactions, TChl acts as one of the main essential parts of energy production in plants, and directly related to the health status of plants based on stress environmental conditions, RWC is a useful indicator for the performance of cell protoplasmic permeability, and intracellular pH regulation is required for trafficking network of proteins and transporting small molecule such as hormones [33,34]. In order to measure the degree of susceptibility to air pollution in each tree species, three fully mature leaves were randomly selected and collected. Leaf samples were immediately put in a plastic bag and stored separately in an icebox or in liquid nitrogen. The analysis of each parameter for APTI was performed as previously described [6,33-36]. APTI values were calculated using the following formula:

APTI = (A × (T + P) + R) / 10

where A is the AsA (mg·g-1 FW, [35]), T is the TChl (mg·g-1 FW, [6]), P is the leaf extract pH [34], and R is the RWC (%, [36]).

Review Report: maybe just "Comparison of PM deposited on leaves".

Author's Reply: (line 234) This graph represents data that is particle concentrations on PM pollutants including TSP, PM10, PM2.5, and PM1.0 at urban forests and roadsides over a three-month field study (and therefore not a comparison of PM particles deposited on leaf surfaces).

Review Report: I suggest to indicate also in the figure captions the long name of the two sites together with their acronyms.

Author's Reply: (lines 235, 243, 264, 288) According to your suggestion, in the captions of Figures 2-4, we changed ‘(a) SFP and (b) YCF’ to ‘(a) Seoul Forest Park (SFP-IN, SFP-OUT) and (b) Yangjae Citizen's Forest (YCF-IN, YCF-OUT)’.

Review Report: ‘to’

Author's Reply: (lines 255-256) We changed ‘~’ to ‘to’.

Review Report: In Figure 4, G. biloba had the lowest capture capacity, compared to the other species. I think you should mention it in the text.

Author's Reply: Thank you for pointing this out. We changed the sentence in the ‘Results’ section.

‘Results’ section (lines 253-256)

3.3. PM adsorption based on LAI and TLA of different tree species

The amount of PM adsorption (g·m-2) in both urban forests and roadsides recalculated based on the LAI is shown in Figure 4. Based on PM adsorption calculated by LAI, Z. serrata had the highest PM capture capacity, while G. biloba had the lowest capture capacity, compared to the other species (one-way ANOVA, p < 0.001), i.e. Z. serrata (1.4 to 3.0 g·m-2), P. occidentalis (0.8 g·m-2), P. densiflora (0.4 to 1.2 g·m-2), P. yedoensis (0.3 to 0.7 g·m-2), and G. biloba (0.1 to 0.4 g·m-2).

Review Report: delete

Author's Reply: (line 295) We have deleted ‘in leaf micromorphology’ in the sentence.

Review Report: RWC indicates the water status of the plant with respect to a full turgidity. Explain better why it should suggest a better tolerance against pollutants.

Author's Reply: We corrected and rewrote ‘Discussion’ section as follows:

‘Discussion’ section (lines 329-338)

The higher RWC values acts as an indicator useful of the plant protoplasmic permeability, indicating possibly more tolerant to environmental pollutions. Air pollutants increases cell permeability, which eventually causes water and nutrient losses, resulting in early leaf senescence [33,34]. Thus, by enhancing RWC positively regulates their variability to regulate the physiological balance of plants under stress conditions [6]. Therefore, higher leaf RWC values are possibly more tolerant to pollutants. As shown in Figure 5, the adsorption and deposition of PM particles was observed around stomatal pores on abaxial leaf surfaces. If the leaf transpiration interrupts due to stomatal limitation by air pollutants, plants can lead to an inhibition of their photosynthesis and growth due to loss of its engine that pulls water up from the soil into the roots.

Review Report: Indicate the references

Author's Reply: We represented the references about previous studies in the ‘Discussion’ section.

‘Discussion’ section (lines 344-346)

Remarkably, the effect of PM adsorption on a per ULA basis was lower in P. occidentalis, P. densiflora, and P. yedoensis as compared to Z. serrata. The current issues and future challenges regarding airborne particulate pollutants depend largely on the tolerance and susceptibility of plants to air pollution [38]. At the physiological and biochemical level, the response of plants to air pollutants can be understood by analyzing the factors that determine tolerance and susceptibility. Previous studies have shown that ascorbic acid content, total chlorophyll content, leaf extract pH, and leaf relative water content are biochemical variables related to tolerance and susceptibility to air pollutants [6,33,34].

Reviewer 2 Report

I find the manuscript too speculative. The authors don't show any quantitative data on leaf morphology. There are only some SEM photos on leaf surface characteristics in Fig. 5. The discussion is also speculative and does not focus on the results. Including APTI seems irrelevant, i.e. does not give any additional value. Morever, the study is not an experimental study. It is a field study.

Author Response

Review Report: I find the manuscript too speculative. The authors don't show any quantitative data on leaf morphology. There are only some SEM photos on leaf surface characteristics in Fig. 5. The discussion is also speculative and does not focus on the results. Including APTI seems irrelevant, i.e. does not give any additional value. Morever, the study is not an experimental study. It is a field study.

Author's Reply: We are thankful for your comments. In accordance with your suggestion in the Reviewer Report, we corrected and rewrote ‘Abstract’, ‘Introduction’, ‘Materials and methods’, ‘Results’, ‘Discussion’, and ‘Conclusions’ sections including your comments. Please refer to the amended and updated versions highlighted in yellow on the uploaded manuscript.

Review Report: I find the manuscript too speculative. The authors don't show any quantitative data on leaf morphology.

Author's Reply: (lines 169-170) As you commented, we added leaf morphological features in Table 2.

Review Report: There are only some SEM photos on leaf surface characteristics in Fig. 5.

Author's Reply: (lines 294-315) We changed and rewrote Figure 5 of ‘Results’ sections.

There were clear differences in the topology of leaf abaxial and adaxial surfaces on five tree species (Figure 5). Airborne particulate pollutants typically accumulated around grooves and epidermal trichomes of adaxial surfaces (especially serrata) and embedded in stomata of abaxial surfaces, blocking the entrance of the stomata (Figure 5a-c). In the case of a detailed survey of epidermal trichomes, the leaf micromorphology of Z. serrata was noticeably more complex on the adaxial surfaces owing to their hairy trichomes (Figure 5a). Both Z. serrata (Figure 5a and 5c) and P. yedoensis (Figure 5d and 5f) showed prominently higher levels of PM particles on the adaxial surfaces. In general, the larger leaf roughness was observed on the leaf adaxial surfaces (especially P. yedoensis, Figure 5f). Moreover, P. occidentalis adsorbed PM particles on grooves with slightly rough surfaces on the adaxial surfaces (Figure 5g and 5i). Interestingly, G. biloba markedly embedded airborne PM particles inside the surrounding stomata of the abaxial surfaces (Figure 5k), and there were a few particulate pollutants on the sparsely arranged grooves of the adaxial sides (Figure 5j and 5l). On the other hand, P. densiflora typically appeared to have the ability to accumulate high airborne particulate particles by exhibiting wax crystals with a dense arrangement in surrounding leaf abaxial and adaxial surfaces (Figure 5m-o).

Review Report: The discussion is also speculative and does not focus on the results. Including APTI seems irrelevant, i.e. does not give any additional value. Moreover, the study is not an experimental study. It is a field study.

Author's Reply: As shown in Figure 2, airborne PM particles were higher outside roadsides than inside urban forests. The main reason for presenting APTI data in this study was used to identify individual differences of APTI which can confirm the air pollution tolerant traits on individual tree species planted in both roadsides and urban forests. The present study revealed that APTI data showed statistically significant higher values in Zelkova serrata planted along roadsides than those of urban forests. These data can help enormously in the establishment of an effective strategy for air pollution management including PM particles.

Author's Reply: We corrected and rewrote ‘Materials and methods’ and ‘Discussion’ sections as follows:

‘Materials and methods’ section (lines 172-180)

2.5. Calculation of air pollution tolerance index (APTI)

APTI was adopted to assess their tolerance level to air pollutants based on the below four parameters. APTI as a marker to evaluate plant tolerance to air pollutants has been evaluated from four physiological and biochemical parameters, i.e. leaf extract pH (pH), relative water content (RWC), total chlorophyll (TChl), and ascorbic acid (AsA) [6,33,34]. For each of these parameters, AsA serve as an important coenzyme in multiple biological metabolism reactions, TChl acts as one of the main essential parts of energy production in plants, and directly related to the health status of plants based on stress environmental conditions, RWC is a useful indicator for the performance of cell protoplasmic permeability, and intracellular pH regulation is required for trafficking network of proteins and transporting small molecule such as hormones [33,34]. In order to measure the degree of susceptibility to air pollution in each tree species, three fully mature leaves were randomly selected and collected. Leaf samples were immediately put in a plastic bag and stored separately in an icebox or in liquid nitrogen. The analysis of each parameter for APTI was performed as previously described [6,33-36]. APTI values were calculated using the following formula:

APTI = (A × (T + P) + R) / 10

where A is the AsA (mg·g-1 FW, [35]), T is the TChl (mg·g-1 FW, [6]), P is the leaf extract pH [34], and R is the RWC (%, [36]).

‘Discussion’ section (lines 328-337)

There were significant variations in the PM adsorption capacities of the different tree species. A temperate deciduous broad-leaved tree (Ulmaceae), Z. serrata, exhibited the highest capacity of 0.06 to 0.11 mg·cm-2 on PM capture per a ULA basis. The effect of leaf area on PM accumulation was observed to be dominant over other investigated characteristics (leaf size and macro- and micro-morphology). As shown in Table 3, the results indicate that P. occidentalis, P. densiflora, and P. yedoensis showed relatively high APTI values, while Z. serrata displayed relatively low the APTI value. The assessment of the above mentioned APTI showed that trees with higher APTI showed the reduction for PM trapping effects and trees with lower APTI revealed a capacity to enhance the capture of air-suspended particles. Consistent with the above the APTI results, Z. serrata revealed particularly distinguishable capacity to prolong APTI values on roadsides as compared to urban forests. Moreover, Z. serrata showed higher RWC on roadsides at two sites, suggesting the presence of better tolerance capacity of plants against air pollutants (Tables S1 and S2). The higher RWC values acts as an indicator useful of the plant protoplasmic permeability, indicating possibly more tolerant to environmental pollutions. Air pollutants increases cell permeability, which eventually causes water and nutrient losses, resulting in early leaf senescence [33,34]. Thus, by enhancing RWC positively regulates their variability to regulate the physiological balance of plants under stress conditions [6]. Therefore, higher leaf RWC values are possibly more tolerant to pollutants. As shown in Figure 5, the adsorption and deposition of PM particles was observed around stomatal pores on abaxial leaf surfaces. If the leaf transpiration interrupts due to stomatal limitation by air pollutants, plants can lead to an inhibition of their photosynthesis and growth due to loss of its engine that pulls water up from the soil into the roots.

Reviewer 3 Report

The topic of the manuscript Understanding the removal efficiencies of individual leaf traits to capture and retain airborne particulate matter is important and fits into the scope of the Forests journal. The results presented are relevant and thought-provoking, which deserves publication. However, the current version of the manuscript requires the following additions. I would like to point out at the issues I am concerned about.

Although the research is conducted thoroughly and provides valuable findings, I am aware of the data on PM concentrations (Table 1), as they are taken from the Seoul Metropolitan Government. It is interesting to compare the numbers with those collected and published by independent researchers. The simple reason for that is a common case of publishing not bias-free data by government bodies. Please give a comment if such a comparison is possible or explain why if it is not. (Although, concentration data on airborne PM were directly obtained in both urban forests and roadsides at each site by the authors, long-term monitoring data are desired too.) Can we apply the proposed framework in ranking species when studying other megacities throughout the world or does it have any restrictions? What are the implementation prospects of the obtained data? Do any of the existing state programs in Seoul give a chance to plant tree species that can maximize the ability to capture and retain PM?

I sincerely hope you will find my suggestions helpful.

Author Response

Review Report: The topic of the manuscript ‘Understanding the removal efficiencies of individual leaf traits to capture and retain airborne particulate matter’ is important and fits into the scope of the Forests journal. The results presented are relevant and thought-provoking, which deserves publication. However, the current version of the manuscript requires the following additions. I would like to point out at the issues I am concerned about.

Although the research is conducted thoroughly and provides valuable findings, I am aware of the data on PM concentrations (Table 1), as they are taken from the Seoul Metropolitan Government. It is interesting to compare the numbers with those collected and published by independent researchers. The simple reason for that is a common case of publishing not bias-free data by government bodies. Please give a comment if such a comparison is possible or explain why if it is not. (Although, concentration data on airborne PM were directly obtained in both urban forests and roadsides at each site by the authors, long-term monitoring data are desired too.) Can we apply the proposed framework in ranking species when studying other megacities throughout the world or does it have any restrictions? What are the implementation prospects of the obtained data? Do any of the existing state programs in Seoul give a chance to plant tree species that can maximize the ability to capture and retain PM?

I sincerely hope you will find my suggestions helpful.

Author's Reply: We are thankful for pointing these comments out. In accordance with your suggestion in the Reviewer Report, we corrected and rewrote ‘Abstract’, ‘Introduction’, ‘Materials and methods’, ‘Results’, ‘Discussion’, and ‘Conclusions’ sections including your comments. Please refer to the amended and updated versions highlighted in yellow on the uploaded manuscript.

Review Report: Although the research is conducted thoroughly and provides valuable findings, I am aware of the data on PM concentrations (Table 1), as they are taken from the Seoul Metropolitan Government.

Author's Reply: (lines 104-108) Table 1 shows the basic data for the selection of similar urban forest sites to averages of airborne PM10 and PM2.5 concentrations in Seoul in this field study.

Review Report: Can we apply the proposed framework in ranking species when studying other megacities throughout the world or does it have any restrictions? What are the implementation prospects of the obtained data?

Author's Reply: Previous studies have suggested that the fine dust adsorption capacity depends on the microstructure of leaves. Plant species, however, vary by country and city, and local atmospheric PM concentrations are also different according to the level of an atmospheric pollutant at a given site. Therefore, it is necessary to rank the species suitable for each region by considering not only the leaf surface micromorphology but also LAI or total leaf area reflecting the canopy properties. According to your suggestion, we corrected and rewrote ‘Discussion’ section as follows:

Discussion (lines 375-384)

Because plants generally exhibit a species‐specificity in associated responses to air pollution, the efficiency for PM adsorption varies across plant species. The leaf micro-morphological features are known to directly affect the agglomeration and capture of fine particles by trees. More specifically, in previous studies that reported a possible association between specific leaf characteristics and PM adsorption, leaf microstructures such as stomata, grooves, and trichomes were found to be optimal areas for capturing PM particles owing to their coarse and adhesive properties [18,46-48]. Notably, our results indicate that the leaf surfaces with high grooves or trichomes have a markedly enhanced ability to retain PM (especially PM2.5) as compared to smooth leaf surfaces. Furthermore, leaf shape and venation of the broad-leaved species have no a significant influence on the immobilization rates and retention times of the fine PM, because individual leaves do not reflect the physical properties of canopy density [18].

On the other hand, canopy density affects air turbulence around leaves, which has been proposed as an important explanatory factor for the deposition of PM particles, enhancing dry deposition of PM on leaves. Several studies have shown that tree species with higher LAI and smaller leaf size were most effective in adsorption of PM [28,29]. LAI can be used as a surrogate for the photosynthetically active area between the atmosphere and underlying land surfaces [30]. In the present study, there was significant interspecies variability of their showing their overall ability to capture airborne particulates when LAI was incorporated for upscaling of unit leaf level to the ecosystem scale. Tree species with higher LAI and smaller leaves were the most effective for PM adsorption. Therefore, the ability to adsorb and retain PM per unit leaf area is important, but it is also important to consider the total leaf area and LAI.

Review Report: Do any of the existing state programs in Seoul give a chance to plant tree species that can maximize the ability to capture and retain PM?

Author's Reply: We corrected and rewrote ‘Conclusions’ sections as follows:

Conclusions (lines 450-457)

Levels of airborne particulates, especially PM, are relatively low in urban forests as compared to that on roadsides, reflecting the removal efficiencies as natural biofilters of urban trees that improve air quality. PM adsorption efficiency by trees under stressful urban conditions is different depending on leaf morphology, physical properties, and microstructure of broadleaved surfaces with well-developed trichomes and grooves. The micro-morphological characteristics such as leaf trichomes and grooves, rather than the macro-morphological characteristics of broadleaved species, can be used as effective indicators for adsorbing PM adsorption. However, their overall ability to capture PM particles by upscaling leaf level to ecosystem-scale especially emphasizes the importance of reflecting the total leaf area and LAI. Regarding the characteristics of leaves, evergreen pine needles may display a higher potential for particulate reduction, especially during winter and early spring, as compared to broad-leaved species. Therefore, as urban open space for green infrastructure programs is limited, if plants are to be used as a means to improve air quality, further studies are needed on various mitigation techniques to maximize airborne PM uptake by trees. Establishing effective policies and management controls to reduce air pollution of particulates is one of the policy tasks of the Korean government. In addition, the Seoul Metropolitan Government and the Korea Forest Service are conducting projects and research for the selection of suitable street trees and urban trees in the construction of urban forests and establish effective policies to reduce particulate matter. This study is a field study conducted as part of green infrastructure programs. The Seoul Metropolitan Government is trying to replace street trees which are able to maximize the ability to capture PM based on the scientific research data.
